# Association of Sentinel Node Biopsy and Pathological Report Completeness with Survival Benefit for Cutaneous Melanoma and Factors Influencing Their Different Uses in European Populations

**DOI:** 10.3390/cancers14184379

**Published:** 2022-09-08

**Authors:** Milena Sant, Maria Chiara Magri, Andrea Maurichi, Roberto Lillini, Maria José Bento, Eva Ardanaz, Marcela Guevara, Kaire Innos, Rafael Marcos-Gragera, Jordi Rubio-Casadevall, Maria-José Sánchez Pérez, Rosario Tumino, Massimo Rugge, Pamela Minicozzi

**Affiliations:** 1Analytical Epidemiology and Health Impact Unit, Department of Research, Fondazione IRCCS Istituto Nazionale dei Tumori, 20133 Milan, Italy; milena.sant@istitutotumori.mi.it (M.S.); mariachiara.magri@istitutotumori.mi.it (M.C.M.); pamela.minicozzi@lshtm.ac.uk (P.M.); 2Melanoma and Sarcoma Surgical Unit, Foundation IRCCS National Cancer Institute, 20133 Milan, Italy; andrea.maurichi@istitutotumori.mi.it; 3North Region Cancer Registry of Portugal (RORENO), Cancer Epidemiology Group, IPO Porto Research Center, 4200-072 Porto, Portugal; mjbento@ipoporto.min-saude.pt; 4Department of Population Studies, Institute of Biomedical Sciences Abel Salazar, University of Porto, 4200-465 Porto, Portugal; 5Navarre Public Health Institute, 31006 Pamplona, Spain; me.ardanaz.aicua@navarra.es (E.A.); mp.guevara.eslava@navarra.es (M.G.); 6Consortium for Biomedical Research in Epidemiology and Public Health (CIBERESP), Institute of Health Carlos III, 28029 Madrid, Spain; rmarcos@iconcologia.net (R.M.-G.); mariajose.sanchez.easp@juntadeandalucia.es (M.-J.S.P.); 7Navarre Institute for Health Research (IdiSNA), 31006 Pamplona, Spain; 8Department of Epidemiology and Biostatistics, National Institute for Health Development, 11619 Tallinn, Estonia; kaire.innos@tai.ee; 9Epidemiology Unit and Girona Cancer Registry, Oncology Coordination Plan, Department of Health, Autonomous Government of Catalonia, Catalan Institute of Oncology (ICO), Girona Biomedical Research Institute (IdiBGi), 08908 Girona, Spain; jrubio@iconcologia.net; 10Andalusian School of Public Health (EASP), Granada Cancer Registry, 18011 Granada, Spain; 11The Biomedical Research Institute (ibs.GRANADA), 18014 Granada, Spain; 12Department of Preventive Medicine and Public Health, University of Granada, 18011 Granada, Spain; 13Cancer Registry and Histopathology Department, Provincial Health Authority (ASP 7), 97100 Ragusa, Italy; rtumino@tin.it; 14Department of Medicine—DIMED, Surgical Pathology and Cytopathology Unit, Università degli Studi di Padova, 35139 Padova, Italy; massimo.rugge@unipd.it; 15Cancer Survival Group, Department of Non-Communicable Disease Epidemiology, London School of Hygiene and Tropical Medicine, London WC1E 7HT, UK

**Keywords:** cutaneous melanoma, population-based cancer registries, pathology report, sentinel lymph node biopsy, relative survival, excess of relative risk of death

## Abstract

**Simple Summary:**

The study was aimed to investigate the frequency of accurate pathology report and sentinel lymph node biopsy for staging clinically node-negative >1 mm melanomas across European countries, which are standard care indicators having relevant consequences for survival. 4245 melanoma cases from in six European countries in 2009–2013 were analyzed by multivariable logistic regression in order to estimate the odds ratio of having such indicators performed. Model-based survival to estimate the five-year relative excess risks of death were computed. Results showed how much accurate pathology profiling and sentinel lymph node biopsy carried survival benefit. Narrowing down between-countries differences in adhesion to guidelines might achieve better outcomes.

**Abstract:**

**Objectives:** Standard care for cutaneous melanoma includes an accurate pathology report (PR) and sentinel lymph node biopsy (SLNB) for staging clinically node-negative >1 mm melanomas. We aimed to investigate the frequency of these indicators across European countries, also assessing consequences for survival. **Methods:** We analyzed 4245 melanoma cases diagnosed in six European countries in 2009–2013. Multivariable logistic regression was used to estimate the Odds Ratio (OR) of receiving complete PR with eight items or SLNB and model-based survival to estimate the five-year relative excess risks of death (RER). **Results:** Overall, 12% patients received a complete PR (range 2.3%, Estonia—20.1%, Italy); SLNB was performed for 68.8% of those with cN0cM0 stage (range 54.4%, Spain—81.7%, Portugal). The adjusted OR of receiving a complete PR was lower than the mean in Estonia (OR 0.11 (0.06–0.18)) and higher in Italy (OR 6.39 (4.90–8.34)) and Portugal (OR 1.39 (1.02–1.89)); it was higher for patients operated on in specialized than general hospitals (OR 1.42 (1.08–1.42)). In the multivariate models adjusted for age, sex, country and clinical-pathological characteristics, the RER resulted in being higher than the reference for patients not receiving a complete PR with eight items (RER 1.72 (1.08–2.72)), or for those not undergoing SLNB (RER 1.76 (1.26–2.47)) Patients with non-metastatic node-negative thickness >1 mm melanoma who did not undergo SLNB had a higher risk of death (RER (RER 1.69 (1.02–2.80)) than those who did. **Conclusions**: Accurate pathology profiling and SLNB carried survival benefit. Narrowing down between-countries differences in adhesion to guidelines might achieve better outcomes.

## 1. Introduction

The quality and completeness of pathology reports (PR), with full descriptions of key parameters, is considered an indicator of standard care [1,2,3,4], as it is important to accurately profile and stage patients, guiding the selection of appropriate treatment and consequently improving the quality of care and outcomes. Despite recent updates in guidelines [5,6], the features currently considered of primary importance in melanoma care and outcomes largely correspond to those used in 2009–2013.

Sentinel Lymph Node Biopsy (SLNB) is recommended as a staging procedure for clinically node-negative cutaneous melanoma of Breslow thickness >1 mm or, on an individual basis, for thinner melanoma in patients with ulceration, high mitotic index (MI), or lympho-vascular invasion [1,7,8] However, the utility of SLNB in controlling distant metastases and improving survival is still debated [9,10].

Differences in the use of SLNB for melanoma evidenced by national population-based studies [11,12,13] can be partially attributable to factors such as the distribution of stage, anatomic location, age at diagnosis, comorbidities, clinicians’ expertise, the health system organization, or may also reflect the lack of solid evidence of benefit for this procedure [10].

The European High Resolution studies (http://www.hrstudies.it/, accessed on 4 July 2022) on samples of cancer cases archived in European population-based cancer registries (CRs) collect more clinical information than is routinely provided by population CRs, according to standardized protocols. Using these data, we aimed to investigate in a real-world context:

(i) the frequency of PR completeness and SLNB use across several European countries, in relation with patients’ and tumor characteristics.

(ii) the impact of PR completeness and SLNB on five-year survival, adjusted by clinic-pathological characteristics, demographic factors and comorbidity.

## 2. Methods

The High Resolution study protocol asked participating CRs to provide at least 300 malignant cutaneous melanoma adult (aged 15 years or more) cases, classified according to the International Classification of Diseases for Oncology, third revision (ICD-O-3) [14], with the morphology code 8720–8790 and topography code C44.0–44.9. Cases had to be diagnosed in 2009–2013 (latest years available), followed up at least to 31 December 2014 and had to include specified information from the clinical records of each case. Trained CR personnel accessed the clinical records and abstracted the relevant information envisaged by the study protocol.

Most CRs provided all the incidences of cases in one or more years of the study period. Registries covering large areas sampled cases from a defined incidence period using a randomized procedure. Appendix A shows the criteria for sampling the cases. The process of identifying eligible cases for the analyses is shown in Figure 1.

We analyzed 4245 operated melanoma cases. Eight countries, with either national (Bulgaria, Estonia) or regional cancer registries (Italy 4 CRs; Portugal 2 CRs; Spain 3 CRs; Switzerland 1 CR), contributed data.

Age at diagnosis was classified as 15–54, 55–64, 65–74, and 75+ years. A score from 1 to 6 was assigned to each Charlson Comorbidity Index (CCI) item [15], and the total was calculated as the sum of the scores for the 19 items. The sum was then rated as 0 (no comorbidities), or ≥1 (one or more).

The anatomical site and morphology of the primary melanoma were coded according to ICD-O3. Topography codes were grouped as the head and neck (ICD-O-3 code C44.0–C44.4), trunk (C44.5), upper limb and shoulder (C44.6), lower limb and hip (C44.7), and unspecified and overlapping regions (C44.8–9). Melanoma morphology was grouped in six subgroups: nodular melanoma (ICD-O-3 code 8721); lentigo malignant melanoma (8741; 8742); superficial spreading melanoma (8743); acral lentiginous melanoma (8744); other types (8722; 8730; 8740; 8744; 8745; 8761; 8770; 8771; 8772); not otherwise specified (NOS) (8720; 8723).

Stage at diagnosis was classified according to the TNM classification, 7th edition [16] and grouped as categories I-IV, or unknown. Tumor thickness was categorized as ≤1 mm, 1.01 mm–2.00 mm, 2.01 mm–4.00 mm, >4.00 mm or unknown.

The Sentinel Lymph Node Biopsy (SLNB) was coded as done, not done and unknown. The following eight items in the PR were considered to be indicative of completeness: melanoma thickness (in millimeters), ulceration (present; absent; unknown), histological subtype, mitotic rate index (0 mitoses per mm^2^; ≥1 mitoses per mm^2^; not mentioned/not available in PR), growth phase (vertical; horizontal; mixed (vertical and horizontal); unknown), lymphocyte infiltration (absent; present, with or without brisk; unknown), tumor regression, and vascular or neural involvement.

A score from 1 to 6 was assigned to each Charlson Comorbidity Index (CCI) item, and the total was calculated as the sum of the scores for the 19 items. The sum was then rated as 0–1 points (no comorbiditiy), ≥1 points (presence of comorbidity) or unknown.

### Statistical Analysis

Multivariable logistic regression was used to establish the roles of different covariates on the PR completeness (versus not complete) and SLNB (compared to not done). Countries’ odds ratios (OR with 95% confidence intervals (CI)) were based on the differences from the balanced grand mean; the common reference for the areas is therefore their grand mean [17].

Relative survival (RS) was calculated as the ratio of the observed survival and the expected survival in the general underlying population. We estimated expected survival by the Ederer II method [18] using CR population life tables stratified by sex, age and the year of diagnosis.

The Relative Excess rate of Risk of death (RER) 5 years after diagnosis, with 95% CI, was estimated with generalized linear models, using 5-year relative survival (RS) as the dependent variable and the other variables under study as covariates [19].

The Akaike Information Criterion (AIC) score was used to determine which models fitted the data best, in order to select the most appropriate variables [20].

Data was analyzed with Stata software, version 14 (StataCorp. 2015. *Stata Statistical Software: Release 14*. College Station, TX: StataCorp LP.)

The study was approved by all participating CRs and by the Institutional Ethical Committee Board of the leading study Institution.

## 3. Results

We analyzed 4245 operated melanoma cases. Bulgaria contributed the fewest number of cases (6.9% of all cases); Italy and Spain had respectively 34.9 and 21.8% of all cases. Patients aged 15–54 years represented 38% of all cases and more than 40% of cases were older than 65 years (Table 1).

The trunk was the most frequent anatomic location in all countries except for Portugal. Superficial spreading melanoma was the most frequent subtype, followed by not-otherwise specified melanoma (NOS). In all the countries, surgery was done mostly in specialized (university or teaching) hospitals (overall 51.6%), or general hospitals (24.2%); only 9.4% were treated as outpatients, or the place of surgery was not specified (14.8%).

Considering the four countries where information on PR was available, overall, 15.2% patients received a complete PR with eight items (range 2.3%, Estonia—20.1%, Italy); in the same countries, SLNB was carried out for 42.6% of the total operated patients, more frequently in Portugal (57.9%), than in Estonia (45.2%), Spain (36.9%) and Italy (38.0%). In these countries, considering the total 795 clinically non-metastatic negative node (cN0M0) and Breslow thickness >1 mm melanoma patients, for whom the procedure is recommended, the overall figure was 68.8%, an average of 81.7% (Portugal), 69.1% (Italy), 65.7% (Estonia) and 54.4% (Spain).

Overall, more than half the patients (54.5%) had TNM I stage at diagnosis, and 3% had TNM IV, but there were notable differences in the stage distribution across countries, e.g., Bulgaria showed the lowest percentage of TNM I and the highest percentage of stage IV, Switzerland had the highest percentage of cases with unknown or incomplete TNM stage; information on comorbidities was available for Estonia, Italy, Portugal and Spain. In these countries, the majority (68.4%) of patients had no comorbidity at diagnosis, while 32% had at least one comorbidity.

Table 2 shows the number of cases receiving a complete pathological report (PR) with eight items, and the adjusted OR with 95% CI, in the four countries providing the relevant information.

This multivariable regression model, adjusted by country, age, sex, stage, place of treatment and CCI, evidenced that, with reference to the mean, the odds of PR completeness were significantly lower in Estonia (OR 0.11 (0.06–0.18)) and higher in Italy (OR 6.39 (4.90–8.34)) and Portugal (OR 1.39 (1.02–1.89)). The odds for PR completeness was higher for patients operated on in specialized rather than general hospitals and for patients with incomplete data on staging (OR 1.74 (1.06–2.87)) than for those with stage I; the lower than reference OR for stage IV (OR 0.22 (0.008–0.61)) was based on five cases only.

The multivariable analysis carried out to investigate whether PR completeness with eight items was associated with survival, showed that, after adjustment by country, age, sex, and stage at diagnosis, patients not receiving a complete PR with eight items had a higher RER of death than those receiving it (RER 2.38 (1.41–4.02)). Appendix A shows the coefficients of each variable included in the model. Of note, a less complete PR, i.e., with four items, was not associated with RER.

Table 3 shows, for the 2272 cases with available information on SLNB, the results of the multivariable analysis carried out to estimate the odds of receiving SLNB, adjusted by country, age, TNM stage, place of treatment and comorbidity.

For each covariate, the table shows the number of cases receiving SLNB and the adjusted OR of receiving this procedure, with 95% CI. With reference to the mean of the pooled countries, the adjusted OR for SLNB was significantly higher in Portugal (OR 1.85 (1.56–2.20)) and lower in Spain (OR 0.82 (0.70–0.95)) and was borderline lower in Italy; with reference to patients aged ≥75, all the younger age classes showed significantly higher ORs of receiving SLNB.

When compared to patients with stage I, those with stage II and III at diagnosis were significantly more likely to receive SLNB (OR 3.99 (3.15–5.10) and 7.18 (5.26–9.79), respectively); the OR was lower than the reference for patients whose stage at diagnosis data were incomplete (OR 0.38 (0.21–0.69)). The odds of receiving SLNB was higher for patients operated on in specialized centers rather than in general hospitals (OR 1.86 (1.50–2.30)).

The multivariable survival analysis (Table 4) including all the 2270 cases with available information on SLNB showed that, by adjusting by the clinic pathological covariates in the model, no significant RER differences across countries were evident.

Patients who did not undergo SLNB had a significantly higher risk of death (RER 1.61 (1.20–2.15)) than those for whom a biopsy was taken. Melanoma pathological features and stage at diagnosis were independent predictors of survival: with reference to thickness >4 mm, the RER of patients with thinner lesions resulted in being significantly lower; ulceration (RER 1.94 (1.38–2.74)) and high or not-mentioned MI (RER 4.22 (1.43–12.44), 3.32 (1.10–10.03)) were independently associated with a higher than reference risk of death. Patients with N0 stage at diagnosis had a lower RER than those with nodal metastases (RER 0.57 (0.40–0.81)) and those with distant metastases at diagnosis had a higher risk (RER 5.24 (2.54–10.83)) than those with no distant metastases.

A model with the same covariates, fitted on the 810 patients with non-metastatic clinically node-negative (cN0M0) thickness >1 mm melanoma, showed that patients who did not undergo SLNB had a higher risk of death (RER 1.69 (1.02–2.80)) than those who did (Figure 2).

For patients with melanoma thickness between 1 mm and 4 mm, the figures were 89.6 (Confidence Interval—CI: 83.5–94.4) vs. 80.8 (CI: 65.0–98.7), respectively; for patients with thickness >4 mm, the figures were 85.8 (CI: 58.4–97.1) vs. 70.3 (CI: 37.6–93.6), respectively.

## 4. Discussion

We found remarkable differences between countries in their adhesion to clinical guidelines on the completeness of PR and SLNB execution for cutaneous melanoma diagnosed in 2009–2013 in Europe. In the multivariable analyses adjusting for potential confounders both these indicators were associated with a five-year risk of death, suggesting that adherence to clinical guidelines can improve disease outcomes.

PR for cutaneous melanoma documents features that are relevant not only in staging and clinical management but are a requisite for planning therapy—particularly with a view to personalized treatment. [1,2,3] Adequately documented PR can support favorable outcomes [21]. We did in fact find that melanoma patients with a well-documented PR had a lower mortality than those with an incomplete, or less complete PR.

The across country differences in the completeness of PR highlighted by our study are in line with results of other studies reporting variation in the compliance with guidelines—especially for differences across countries—as observed in the International Collaboration on Cancer Reporting panel (ICCR) [22] and other studies [23,24,25,26]. Geographic differences in the availability of pathological information may also depend on the availability and accuracy of the PR themselves. Therefore, adherence to international evidence-based protocols yielding more complete PRs is to be encouraged and remains a main goal in recent clinical procedures [21,24].

The higher odds of receiving complete PR and SLNB in specialized oncologic centers than in general hospitals points to the usefulness of centralizing the management of melanoma patients [27]. The lack of association of comorbidity with SLNB performance or PR completeness, once the place of treatment (and other factors) was adjusted for in the multivariable analysis, suggests that specialized oncologic facilities may provide better melanoma management than others, independently from the presence of comorbid conditions.

In our study, SLNB was performed for around 43% of total cases, with notable differences between countries, however, by restricting the analysis to patients with non-metastatic clinically node negative melanoma (cN0M0) with a tumor thickness > 1 mm, for whom the procedure is recommended, the percentage of biopsies rose to almost 70% (69.8%), a figure close to that reported in the US Surveillance Epidemiology and End Results (SEER) in a comparable study period [23]. Although the intercountry variability in the SLNB use attenuated when considering clinically node-negative cases only, the multivariable analysis confirmed the differences by country, as well as the lower odds of receiving SLNB for elderly than younger patients, a finding reported also by other studies [11,28].

Our results are consistent with those of other studies reporting slightly higher than European mean SLNB frequencies in Spain [11] and lower ones in Italy [29]; in both these countries, a remarkable within-country variation in SLNB use was highlighted. In the present study, the large proportion of early-stage melanomas in Spain (Table 1) might explain the overall low odds of SLNB, as well as an RER in line with the European average. Other nationwide registrations or dedicated quality registries on melanomas detected within-country differences: for instance, during 2005–13 in the Netherlands, SLNB was performed for 50% of non-distant metastatic melanoma patients on average, ranging from 22.5 to 56.5% across regions [12] The German national melanoma registry reported a higher percentage of SLNB (82%), but did not look into regional differences [13].

We noted that in our population-based real-world setting, SLNB was done not only for patients with intermediate thickness or thick clinically node-negative non distant metastatic melanoma, for whom the benefit of this procedure has been proven [8,30], but also for a proportion of patients with thinner or unknown-thickness lesions, or patients with uncertain nodal status. Underreporting and incompleteness of clinical documentation might have prevented a more precise definition of tumor stage for these patients. Also, it cannot be excluded that some of them presented clinical indications to SLNB that were not captured by our study.

The survival benefit carried by SLNB for intermediate-thickness and thick melanomas found by our study is in line with population- [28] and hospital-based studies [31]. The finding that in comparison with thick melanoma the RER carried by SLNB decreased also for thin melanoma is consistent with recent studies indicating SLNB should be considered for selected high-risk patients, defined by features such as a high MI, ulceration, lympho-vascular invasion, tumor-infiltrating lymphocytes, or regression [31,32,33].

Lymph node dissection could have improved the survival of cases with nodal metastases. In our study, considering patients with stage II-IV, the inclusion of nodal dissection execution in a multivariable analysis adjusted by all the above clinical-pathological factors did not evidence a statistical significant effect of this surgery on the risk of death (RER of patients undergoing lymph-node dissection 0.67, 95% CI 0.33–1.34).

Past population-based studies have evidenced higher survival for women than men [34] and for younger than older patients [35,36], as well as remarkable across-European-country survival inequalities [37]. In the present study, adjustment for stage at diagnosis and pathological features, such as MI, ulceration, thickness, explained the lower mortality of women and younger ages, as well as geographic differences.

The presence of comorbid conditions may cause a delay in diagnosis or contraindicate intensive treatments, thereby decreasing survival. In contrast with other studies, in our dataset comorbidity resulted in not being independently associated with the RER. In a recent population-based study the detrimental effect of comorbidity on melanoma survival was concentrated in patients with an advanced tumor stage at diagnosis [38]. Another study documented that melanoma patients with >2 CCI had a significantly higher risk of death than those with no comorbidity [39]. The lack of statistically significant associations of comorbidity with RER in our study could be attributable to the low number of cases with advanced stage at diagnosis, or with severe comorbidities. In fact, our study population was largely represented by early-stage melanoma (54% TNM stage I) and patients with no comorbidities (68% overall, with similar distribution by stage).

Comorbidity data were abstracted from each patient’s clinical record and their availability and completeness may vary by hospitals and clinician’s attitudes to documenting comorbid conditions in the clinical notes. However, we cannot exclude that, for a certain proportion of cases with comorbidity coded as “absent”, the relevant information was actually unknown.

In all countries, more than 50% of melanoma patients had tumor stage I at diagnosis; a notable exceptions was Bulgaria, where the most frequent stage category was represented by stage II, and the percentage of stage IV was the highest among the included countries. The more advanced tumor stage at diagnosis likely explains the lower-than-European average melanoma survival in this country that was previously reported [40]. The five-year relative survival of Bulgarian patients included in this study was 67% (95% CI 0,54–0.77). In contrast, the relatively low frequency of stage I melanoma in Switzerland (42%) is counterbalanced by the highest percentage of cases with unknown stage at diagnosis (45%) and was not associated with low survival: the 5-year relative survival of Swiss patients in this study was 98% (95 CI 0.129–0.999). Due to a lack of information we could not analyze the association of stage at diagnosis with SLNB or PR in these two countries.

The present study did not focus on treatment. However, patients were diagnosed prior to the use of the new anti BRAF/MEK drugs or immunotherapy as adjuvant treatment, starting in 2018. The use of interferon, the drug previously approved in adjuvant therapy, varied considerably in the different countries [40]. In our study, 90 patients received adjuvant treatment in addition to surgery, of whom 68 received interferon.

In the multivariable analysis adjusted by all clinical pathological factors considered in the study, the administration of adjuvant chemotherapy or target treatment resulted in being not associated with survival (RER of patients receiving adjuvant chemotherapy or target treatment 0.96, 95% CI 0.38–2.46).

A strength of our study was the use of all incidences of cases (or representative samples of them) during the study years in the participating CR areas, irrespective of the type of treatment and hospital, or its location within or outside the CR area. Hence, variations in the completeness and quality of data provided by the hospitals reflect actual variations in current clinical practices. Furthermore, the centralization of data for common checks and analyses ensured uniform methods of analysis and data comparability.

## 5. Conclusions

Despite the existence of clinical recommendations, we found notable across-country differences in Europe in the completeness of PR for cutaneous melanoma, as well as in the use of SLNB. The odds of receiving a complete PR or undergoing SLNB was higher for patients treated in specialized oncologic centers than in general hospitals. Multivariable analysis adjusting for potential confounders suggested that a complete pathological report and SLNB were associated with survival benefit. Narrowing down the differences between countries by adherence to guidelines is important for achieving more favorable outcomes.

## Figures and Tables

**Figure 1 cancers-14-04379-f001:**
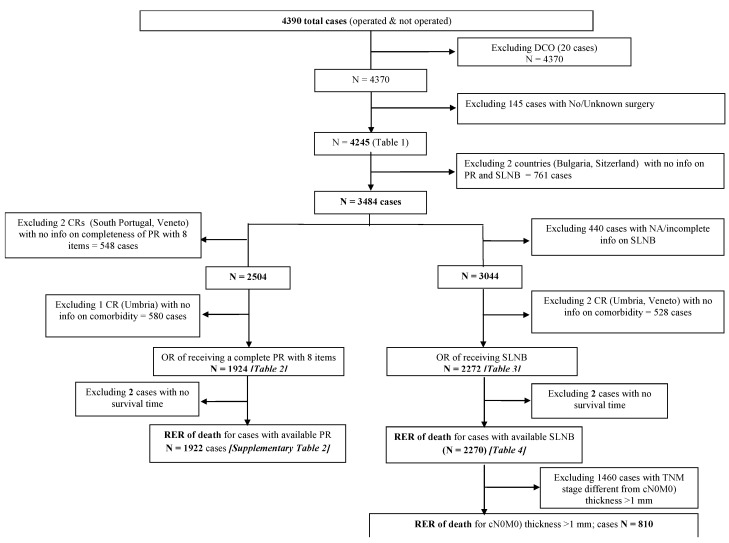
Process of identifying eligible cases for the analyses.

**Figure 2 cancers-14-04379-f002:**
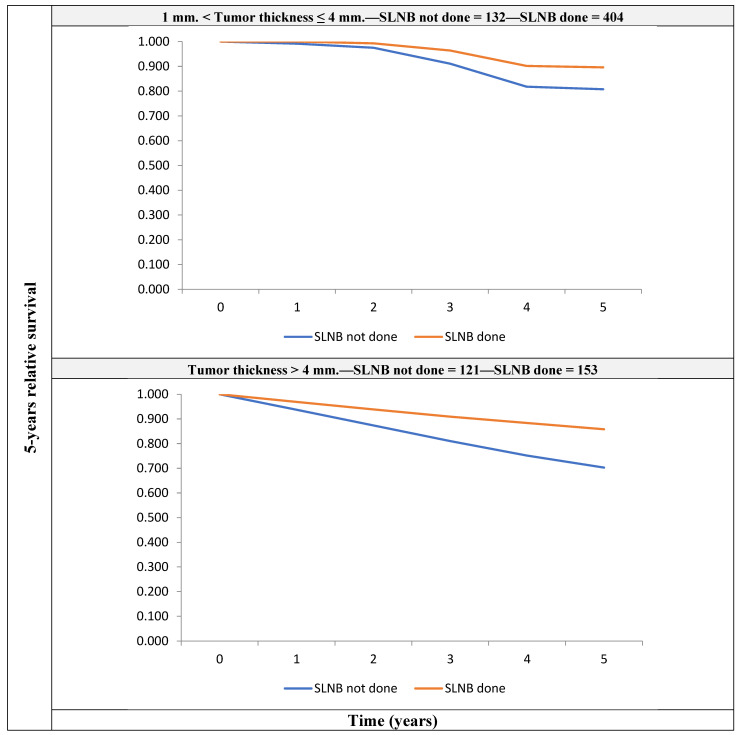
5-year crude relative survival by tumor thickness, for patients with non-metastatic clinically node-negative (cN0M0) melanoma with thickness >1 mm, for those who underwent SLNB and those who did not receive it.

**Table 1 cancers-14-04379-t001:** Distribution and clinical and pathological characteristics of 4245 operated cutaneous melanoma patients diagnosed in 2009–2013 in six European countries.

	All patients	Bulgaria	Estonia	Italy	Portugal	Spain	Switzerland
*N (%)*	*4245 (100.0)*	*295 (6.9)*	*345 (8.1)*	*1483 (34.9)*	*732 (17.2)*	*924 (21.8)*	*466 (10.9)*
**Sex**							
Men	2066 (48.7)	142 (48.1)	137 (39.7)	765 (51.6)	334 (45.6)	451 (48.8)	237 (50.9)
Women	2179 (51.3)	153 (51.9)	208 (60.3)	718 (48.4)	398 (54.4)	473 (51.2)	229 (49.1)
**Age at diagnosis (years)**							
15–54	1621 (38.2)	89 (30.2)	132 (38.3)	607 (40.9)	275 (37.6)	355 (38.4)	163 (35.0)
55–64	801 (18.9)	80 (27.1)	59 (17.1)	263 (17.7)	149 (20.4)	165 (17.9)	85 (18.2)
65–74	856 (20.2)	62 (21)	81 (23.5)	293 (19.8)	148 (20.2)	175 (18.9)	97 (20.8)
≥75	967 (22.8)	64 (21.7)	73 (21.2)	320 (21.6)	160 (21.9)	229 (24.8)	121 (26)
**Anatomical Site**							
Head & neck	624 (14.7)	61 (20.7)	34 (9.8)	161 (10.9)	105 (14.3)	191 (20.7)	72 (15.5)
Trunk	1696 (40)	124 (42)	182 (52.8)	590 (39.8)	248 (33.9)	366 (39.6)	186 (39.9)
Upper limb & shoulder	631 (14.9)	28 (9.5)	49 (14.2)	220 (14.8)	119 (16.3)	125 (13.5)	90 (19.3)
Lower limb & hip	1061 (25)	67 (22.7)	77 (22.3)	337 (22.7)	254 (34.7)	211 (22.8)	115 (24.7)
Skin NOS	233 (5.5)	15 (5.1)	3 (0.9)	175 (11.8)	6 (0.82)	31 (3.35)	3 (0.64)
**Histological subtype ^1^**							
Nodular melanoma	444 (10.5)	65 (22)	18 (5.2)	131 (8.8)	90 (12.3)	114 (12.3)	26 (5.6)
Lentigo maligna	233 (5.5)	4 (1.4)	6 (1.7)	38 (2.6)	26 (3.5)	115 (12.4)	44 (9.44)
Superficial spreading	1698 (40)	21 (7.1)	12 (3.5)	542 (36.6)	307 (41.9)	493 (53.4)	323 (69.3)
Acral lentiginous melanoma	101 (2.4)	1 (0.3)	13 (3.8)	20 (1.3)	32 (4.37)	21 (2.27)	14 (3)
Other	715 (16.8)	15 (5.1)	174 (50.4)	431 (29.1)	23 (3.1)	63 (6.8)	9 (1.9)
NOS	1053 (24.8)	189 (64.1)	122 (35.4)	320 (21.6)	254 (34.7)	118 (12.8)	50 (10.7)
**Place of Primary Surgery**							
Specialized hospital	2189 (51.6)	253 (85.8)	298 (86.4)	781 (52.7)	214 (29.2)	557 (60.3)	86 (18.5)
General hospital	1026 (24.2)	32 (10.8)	25 (7.2)	119 (8)	490 (66.9)	360 (39.0)	0 (0.0)
Outpatient	400 (9.4)	0 (0.0)	15 (4.4)	0 (0.0)	1 (0.14)	6 (0.6)	378 (81.1)
Unknown	630 (14.8)	10 (3.4)	7 (2)	583 (39.3)	27 (3.7)	1 (0.1)	2 (0.4)
**Completeness of pathological report ^2^**							
Complete (8 histopathological items)	528 (15.2)	NA	8 (2.3)	298 (20.1)	58 (7.9)	164 (17.8)	NA
Incomplete (<8 items), unknown, not available	2956 (84.8)	NA	337 (97.7)	1185 (79.9)	674 (92.1)	760 (82.2)	NA
**Sentinel Lymph Node Biopsy ^3^**							
Done	1485 (42.6)	NA	156 (45.2)	564 (38.0)	424 (57.9)	341 (36.9)	NA
Not done	1559 (44.7)	NA	159 (46.1)	555 (37.4)	302 (41.3)	543 (58.8)	NA
Unknown	440 (12.6)	NA	30 (8.7)	364 (24.5)	6 (0.8)	40 (4.3)	NA
**Sentinel Lymph Node Biopsy in cN0cM0 and Thickness > 1 mm**							
Done	557 (68.8)	NA	94 (65.7)	112 (69.1)	228 (81.7)	123 (54.4)	NA
Not done	253 (31.2)	NA	49 (34.3)	50 (30.9)	51 (18.3)	103 (45.6)	NA
**TNM Stage**							
I	2313 (54.5)	78 (26.4)	180 (52.2)	884 (59.6)	380 (51.9)	596 (64.5)	195 (41.8)
II	801 (18.9)	135 (45.8)	101 (29.3)	186 (12.5)	167 (22.8)	180 (19.5)	32 (6.9)
III	483 (11.4)	28 (9.5)	46 (13.3)	150 (10.1)	142 (19.4)	88 (9.5)	29 (6.2)
IV	128 (3.0)	19 (6.4)	7 (2.0)	46 (3.1)	30 (4.1)	26 (2.8)	0 (0.0)
Incomplete or unknown	520 (12.2)	35 (11.9)	11 (3.2)	217 (14.6)	13 (1.8)	34 (3.7)	210 (45.1)
**Charlson comorbidity index (CCI)**							
CCI 0	1490 (68.4)	NA	225 (65.2)	1376 (92.8)	521 (71.2)	593 (64.2)	NA
CCI 1 or higher	687 (31.9)	NA	120 (34.8)	104 (7.0)	211 (28.8)	252 (27.3)	NA
Unknown	82 (8.9)	NA	0 (0.0)	3 (0.2)	0 (0.0)	79 (8.6)	NA

Notes: ^1^ Histological subtype definition (according to ICDO-3 morphological codes): 8720; Nodular melanoma: 8721; Lentigo maligna melanoma: 8741–8742; Superficial spreading melanoma: 8743; Acral lentiginous melanoma: 8744; Other melanoma: 8722; 8723; 8730; 8740; 8745; 8761; 8770; 8771; 8772; Malignant melanoma, NOS: 8720. ^2^ Complete pathological report with eight recommended histopathological items: Breslow thickness, ulceration, histological subtype, mitotic rate, growth phase, lymphocyte infiltration, tumor regression, and vascular or neural involvement. ^3^ Percentages of total operated cases in the 4 countries with available information on SLNB. All characteristics resulted in significantly different values across countries (*p* < 0.05; Chi Squared test).

**Table 2 cancers-14-04379-t002:** Number of cases (*N*) receiving a complete pathological report (PR) with eight recommended histopathological items ^1^ and adjusted Odds Ratios (OR) with 95% Confidence Intervals (CI), for cutaneous melanoma patients diagnosed in 2009–2013 in four European countries.

	*N*	Adj. OR	95% CI	*p*-Value
Total	441			
**Country ^2^**				
Estonia	8	**0.11**	[0.06–0.18]	0.000
Italy	211	**6.39**	[4.90–8.34]	0.000
Portugal	58	**1.39**	[1.02–1.89]	0.038
Spain	164	**1.06**	[0.83–1.35]	0.642
**Age at diagnosis (years)**				
15–54	185	**1.21**	[0.85–1.71]	0.286
55–64	85	**1.39**	[0.94–2.06]	0.096
65–74	78	**1.06**	[0.73–1.56]	0.752
75+	93	**1**		
Sex				
Men	209	**1**		
Women	232	**0.97**	[0.76–1.24]	0.808
**Stage at diagnosis**				
I	223	**1**		
II	83	**1.01**	[0.73–1.40]	0.937
III	52	**0.83**	[0.56–1.22]	0.336
IV	5	**0.22**	[0.08–0.61]	0.003
Incomplete	66	**1.74**	[1.06–2.87]	0.029
Unknown	12	**0.94**	[0.43–2.05]	0.885
**Place of Primary surgery**				
General hospital ^3^	143	**1**		
Specialized hospital	298	**1.42**	[1.08–1.86]	0.012
**Charlson comorbidity index (CCI)**				
CCI 0	300	**1**		
CCI ≥ 1	124	**1.06**	[0.79–1.42]	0.700
Unknown	17	**1.09**	[0.58–1.48]	0.749

Notes: ^1^ Eight recommended histopathological items: Breslow thickness, ulceration, histological subtype, mitotic rate, growth phase, lymphocyte infiltration, tumor regression, and vascular or neural involvement. ^2^ Reference for country is the European mean. ^3^ Including surgery in general hospitals, as outpatients or in not specified places.

**Table 3 cancers-14-04379-t003:** Number of cases (*N*) receiving Sentinel Lymph Node Biopsy and adjusted Odds Ratios (OR) with 95% Confidence Intervals (CI), for cutaneous melanoma patients diagnosed in 2009–2013 in four European countries.

	*N*	Adj. OR	95% CI	*p*-Value
Total	1054			
**Country ^1^**				
Estonia	156	**0.83**	[0.67–1.02]	0.078
Italy	133	**0.8**	[0.65–0.99]	0.041
Portugal	424	**1.85**	[1.56–2.20]	0.000
Spain	341	**0.82**	[0.70–0.95]	0.008
**Age at diagnosis (years)**				
15–54	444	**3.3**	[249–4.36]	0.000
55–64	220	**3.03**	[2.23–4.13]	0.000
65–74	224	**2.25**	[1.68–3.02]	0.000
75+	166	**1**		
**Sex**				
Men	488	**1**		
Women	566	**1.10**	[0.92–1.33]	0.302
**Stage at diagnosis**				
I	449	**1**		
II	307	**3.99**	[3.13–5.10]	0.000
III	251	**7.18**	[5.26–9.79]	0.000
IV	23	**1.10**	[0.64–1.88]	0.727
Incomplete	17	**0.38**	[0.21–0.69]	0.001
Unknown	7	**1.01**	[0.41–2.51]	0.977
**Place of Primary surgery**				
General hospital ^3^	424	**1**		
Specialized hospital	630	**1.86**	[1.50–2.30]	0.000
**Charlson comorbidity index (CCI)**				
CCI 0	756	**1**		
CCI ≥ 1	289	**0.83**	[0.67–1.04]	0.109
Unknown	9	**0.25**	[0.11–0.56]	0.001

Notes: ^1^ Reference for country is the European mean. ^3^ Including surgery in general hospitals, as outpatients or in not specified places.

**Table 4 cancers-14-04379-t004:** Number of cases and adjusted 5-year Relative Excess rate of Risk of death (RER) with 95% Confidence Intervals (CI) for cutaneous melanoma patients diagnosed in 2009–2013 in four European countries.

	*N*	RER	95% CI	*p*-Value
**Country ^1^**				
Estonia	315	**1.04**	[0.75–1.45]	0.836
Italy	347	**0.94**	[0.63–1.40]	0.773
Portugal	726	**1.09**	[0.86–1.38]	0.488
Spain	882	**0.94**	[0.73–1.21]	0.631
**Age at diagnosis (years)**				
15–54	886	**0.76**	[0.51–1.12]	0.162
55–64	415	**1.03**	[0.68–1.56]	0.902
65–74	452	**1.10**	[0.76–1.60]	0.619
75+	517	**1**		
Sex				
Men	1058	**1**		
Women	1212	**0.82**	[0.62–1.08]	0.160
**Sentinel lymph node biopsy**				
Executed	1054	**1**		
Not Executed	1216	**1.61**	[1.20–2.15]	0.002
**Mitotic index**				
0 mitoses per mm^2^	665	**1**		
≥1 mitoses per mm^2^	987	**4.22**	[1.43–12.44]	0.009
Not mentioned/not available in PR	618	**3.32**	[1.10–10.03]	0.033
**Thickness**				
<= 1 mm	1128	**0.10**	[0.05-.0.22]	0.000
1.01 mm–2 mm	369	**0.20**	[0.11–0.38]	0.000
2.01 mm–4 mm	314	**0.72**	[0.52–0.99]	0.041
> 4 mm	414	**1**		
Unknown	45	**0.55**	[0.18–1.65]	0.287
**Ulceration**				
Absent	579	**1**		
Present	1606	**1.94**	[1.38–2.74]	0.000
Not mentioned in PR	85	**1.78**	[0.63–5.06]	0.279
**Nodal stage**				
N0	1584	**0.57**	[0.40–0.81]	0.002
N1	50	**1.30**	[0.78–2.15]	0.309
N2	30	**1.09**	[0.60–1.98]	0.783
N3	15	**1.02**	[0.43–2.40]	0.968
N+	3	**2.48**	[0.54–11.41]	0.242
Nx	588	**1**		
**M stage**				
M0	1954	**1.02**	[0.54–1.92]	0.957
M1	61	**5.24**	[2.54–10.83]	0.000
Mx	255	**1**		
**Charlson comorbidity index (CCI)**				
CCI 0	1551	**1**		
CCI > = 1	668	**1.26**	[0.94–1.69]	0.116
Unknown	51	**1.41**	[0.51–3.94]	0.512

Notes: ^1^ Reference for country is the European mean.

## Data Availability

Data are not publicly available. They can be obtained by the Melanoma HR Study Working Group after motivated request.

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
