# Peer review of "Association of Sentinel Node Biopsy and Pathological Report Completeness with Survival Benefit for Cutaneous Melanoma and Factors Influencing Their Different Uses in European Populations"

_cancers, 2022, doi:10.3390/cancers14184379_

Round 1

Reviewer 1 Report

A very interesting article for clinical practice. Two important issues were raised - the quality of histopathological reports and the performance of SNLB. Thank you for you contribution.

Reviewer 2 Report

In the the manuscript "Association of sentinel node biopsy and pathological report completeness with survival benefit for cutaneous melanoma and factors influencing their different uses in European populations" by Milena Sant et al. the authors conduct an extremely interesting study on the practices across Europe regarding the handling of cutaneous melanoma patients. The study is extremely relevant and interesting and the paper is well-written and organized. However, there are minor aspects that merit the attention of the authors:

1. The http://www.hrstudies.eu/ website does not work. Only the other  http://www.hrstudies.it/ website works properly. Please correct this aspect.

2. The sentence Sentinel Lymph Node Biopsy (SLNB) was coded as done, not done and unknown. is repeated twice, a few sentences apart. It should be corrected.

3. The legend of figure 2 should be at the bottom of the figure and not at the top.

4. The authors should expand the discussion section to address some of the shortcomings of the study, namely, it is a portray of some selected centers in each country involved and not a picture of the whole country. Besides, there is an important heterogeneity regarding the pathological evaluation of all the cases. In addition, there is no comparison regarding the treatment modalities used for each case, which could have critically impacted on patient mortality.

Thus, it will be mandatory to perform additional changes in the present manuscript for it to be considered suitable for publication in Cancers.
